



**Antarctic atmospheric boundary layer observations with the Small Unmanned**

**Meteorological Observer (SUMO)**

John J. Cassano[1,2], Melissa A. Nigro[2], Mark W. Seefeldt[1], Marwan Katurji[3], Kelly Guinn[1,2], Guy

5     Williams[4], Alice DuVivier[5]

[1] Cooperative Institute for Research in Environmental Sciences, University of Colorado, Boulder,

CO, USA

[2] Department of Atmospheric and Oceanic Sciences, University of Colorado, Boulder, CO, USA

10     [3] Department of Geography, University of Canterbury, Christchurch, New Zealand

[4] Autonomous Maritime Systems Laboratory, University of Tasmania, Launceston, Australia

[5] National Center for Atmospheric Research, Boulder, CO, USA

Correspondence to: John J. Cassano (john.cassano@colorado.edu)



**Abstract**

Between January 2012 and June 2017 a small unmanned aerial system (UAS), known as the Small Unmanned Meteorological Observer (SUMO), was used to observe the state of the atmospheric boundary layer in the Antarctic. During 6 Antarctic field campaigns 116 SUMO flights were completed. These flights took place during all seasons over both permanent ice and ice free locations on the Antarctic continent and over sea ice in the western Ross Sea. Sampling was completed during spiral ascent and descent flight paths that observed the temperature, humidity, pressure and wind up to 1000 m above ground level and sampled the entire depth of the atmospheric boundary layer as well as portions of the free atmosphere above the boundary layer. A wide variety of boundary layer states were observed including very shallow, strongly stable conditions during the Antarctic winter and deep, convective conditions over ice free locations in the summer. The Antarctic atmospheric boundary layer data collected by the SUMO sUAS, described in this paper, can be retrieved from the United States Antarctic Program Data Center (https://www.usap-dc.org). The data for all flights conducted on the continent are available at https://www.usap-dc.org/view/dataset/601054 (Cassano 2017; https://doi.org/10.15784/601054) and data from the Ross Sea flights, are available at https://www.usap-dc.org/view/dataset/601191 (Cassano 2019; https://doi.org/10.15784/601191).



## 1. Introduction

The turbulent lower portion of the atmosphere, known as the atmospheric boundary layer, is the part of the atmosphere that interacts directly with the underlying surface. In lower and middle latitudes atmospheric properties in the boundary layer change diurnally in response to the diurnal cycle of net radiation at the surface. During the day downwelling shortwave radiation often results in a positive surface radiation budget, surface heating, and the development of a convective boundary layer with temperature decreasing with height at a rate of approximately 10 K km$^{-1}$. At night, longwave radiative cooling from the surface results in surface cooling and the development of a statically stable boundary, often characterized by a surface based inversion, where temperature increases with height. The presence of clouds or changes in large-scale winds will alter this typical diurnal boundary layer evolution (Stull 1988).

In the polar regions a weaker, or absent, diurnal cycle in radiative forcing results in a less pronounced diurnal cycle in boundary layer evolution compared to that observed in lower latitudes, although some locations, such as the McMurdo Dry Valleys, do experience a pronounced diurnal cycle during the austral summer (Katurji et al. 2013). The presence of extensive ice covered surfaces reduces the amount of solar radiation absorbed at the surface during the day and weakens, or eliminates, the presence of convective boundary layer conditions. During the long polar night extended periods of radiative cooling at the surface lead to the development of stably stratified boundary layers with strong temperature inversions, although strong winds or clouds can cause the surface inversion to dissipate and well mixed conditions to develop (King and Turner 1997; Cassano et al. 2016a; Nigro et al. 2017).

The vast majority of in-situ atmospheric observations in the Antarctic are surface observations within the lowest 10 m of the atmosphere with very few observations made above the surface (Summerhayes 2008). This lack of information on the vertical structure of the atmosphere, even in the lowest 10s to 100s of meters, limits our ability to study the Antarctic boundary layer. Cassano et al. (2016a) provide a summary of Antarctic boundary layer studies that made in-situ observations of vertical profiles of atmospheric properties. Antarctic field campaigns from the 1950s



to present have often relied on data collected on meteorological towers that extend up to 50 m

above the surface.

More recently several groups have used unmanned aerial systems (UAS) to observe the Antarctic boundary layer (Cassano 2014). The British Antarctic Survey was the first to use UAS for Antarctic boundary layer observations with 20 science flights conducted in October and December 2007 (P. Anderson, personal communication 2013). Cassano et al. (2010, 2016b) and Knuth et al.

(2013) describe UAS flights which observed air-sea exchanges in the Terra Nova Bay polynya in the western Ross Sea. The Finnish Meteorological Institute has conducted UAS flights in Dronning Maud Land, Antarctica, at the Aboa research station. Several different UAS were deployed from the *RV Polarstern* ice breaker in the Weddell Sea during the austral winter of 2013 (Jonassen et al. 2015).

Our research group has used a small, easily deployed UAS (sUAS) known as the Small Unmanned Meteorological Observer (SUMO) during six Antarctic field campaigns from 2012 through 2017. These SUMO campaigns have occurred on the Ross Ice Shelf, near Ross Island, and in the McMurdo Dry Valleys (Fig. 1) in the austral summer and late austral winter into early spring. Austral autumn and winter SUMO flights were conducted as part of the polynyas, ice

production and seasonal evolution in the Ross Sea (PIPERS) research cruise in the western Ross Sea from April to June 2017 (Ackley et al. 2020). The SUMO campaigns were conducted over permanent ice shelves in areas with little topography within 100 or more km as well as in regions of very complex terrain with elevations rising over 4000 m within 30 km of the flight locations. Other flights occurred over the ice free, complex terrain of the Wright Valley and over the sea ice of the

Ross Sea. The data collected over these varied surfaces over the entire annual cycle provides observations of a wide range of Antarctic boundary layer conditions.










Figure 1: Location of all SUMO sUAS boundary layer flights (red squares) over the Antarctic continent (a) and the Ross Sea (b). Maps prepared by Michael Wethington, Polar Geospatial Center, Department of Earth Sciences, University of Minnesota.

This paper describes the SUMO sUAS and flight strategy employed during our Antarctic field campaigns (section 2), the data processing and quality control applied to the data (section 3), provides examples of boundary layer features that were observed with the SUMO sUAS (section 4) and the SUMO data availability (section 5). A brief summary is presented in section 6.

**2. The SUMO sUAS and flight strategy**

**2.1 SUMO sUAS**

Reuder et al. (2012), Cassano (2014), and Jonassen et al. (2015) provide technical descriptions of the SUMO sUAS. The SUMO is a small fixed wing pusher prop drone with 0.80 m wingspan and 580 g take-off weight that is constructed from high-density foam (Fig. 2). The

airframe is based on the commercially available Multiplex Funjet model remote control airplane. The SUMO uses a single lithium polymer (LiPo) battery to power the electric motor, which allows for a flight time of ~30 minutes. Further details about the SUMO sUAS are given in Table 1.



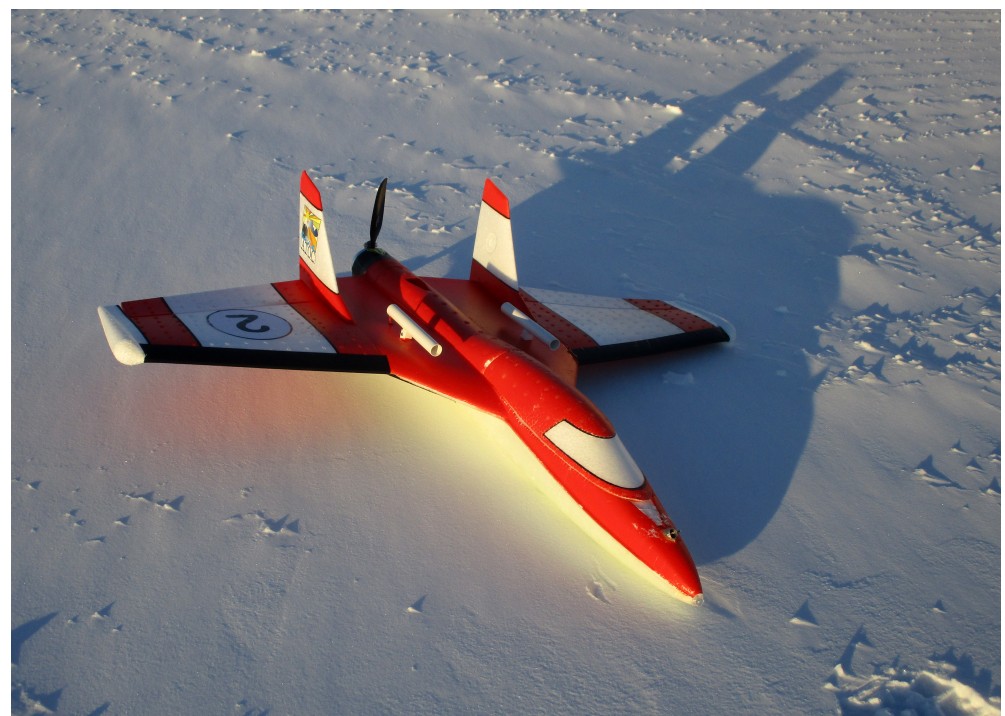

110         Figure 2: SUMO sUAS at the Pegasus ice runway outside of McMurdo Station,

Antarctica.

Table 1: SUMO sUAS airframe and flight specifications (after Reuder et al. 2012 and Cassano

2014).


| | |
|---|---|
| Wingspan | 0.80 m |
| Length | 0.75 m |
| Propeller diameter | 227 mm |
| Take-off weight | 580 g |
| Motor | 120 W electric brushless |
| Battery | 2.1 Ah, 11.1 V lithium polymer |
| Speed (min / cruise / max) | 8 m s$^{-1}$ / 15 m s$^{-1}$ / 42 m s$^{-1}$ |



| | |
|---|---|
| Horizontal range | 5 km |
| Vertical range | 4 km a.g.l. |
| Flight duration | ~30 min |

The SUMO sUAS is launched by hand and lands on its underside. Usually a two-person team operates the SUMO. One person, the remote control pilot, maintains visual sight of the aircraft and controls the SUMO via a remote control while the second flight team member, the ground

station pilot, manages the ground control software. A standard model airplane remote control can be used to manually control the SUMO sUAS but it is typically flown in a semi-autonomous or autonomous mode via the onboard Paparazzi autopilot (ENAC 2008) and ground control software (Table 2). A 2.4 GHz radio modem is used for two-way communication between the SUMO and the ground control software. The SUMO observations are relayed to the ground station computer and

the pre-programmed flight plan can be modified with the ground station software via the radio modem link. In case of a ground station communication failure the remote control pilot can take control of the aircraft at any time.

Table 2: SUMO sUAS navigation, control and communication and scientific instrumentation

sensor range, accuracy, acquisition frequency and time constant (after Reuder et al. 2012 and Cassano 2014).

| Navigation, Control and Communication | |
|---|---|
| Auto-pilot navigation | On-board GPS |
| Manual navigation | Model airplane remote control |
| Attitude control | Diydrones Ardu inertial measurement unit (IMU) |
| Communication | 2.4 GHz two-way data link with Toughbook laptop computer |

| Scientific Instrumentation | | | | | |
|---|---|---|---|---|---|
| Meteorological Parameter | Sensor | Range | Accuracy | Acquisition Frequency | Sensor Time |





| | | | | | Constant |
|---|---|---|---|---|---|
| Pressure | VTI SCP1000 | 300-1200 hPa | n/a | 4 Hz | n/a |
| Temperature | Pt 1000 Heraeus M222 | -32 to 96°C | ±0.2 K | 8 Hz | ~3 s |
| Temperature | Sensirion SHT 75 | -40 to 124°C | ±0.3 K | 2 Hz | 5 to 30 s |
| Humidity | Sensirion SHT 75 | 0-100% | ±2% | 2 Hz | ~8 s |
| Wind | "No flow sensor" algorithm (Mayer et al. 2012) | n/a | Wind speed: ~1 m s$^{-1}$ Wind direction: ~5° | ~30 s | n/a |

The SUMO records temperature, humidity, pressure and aircraft location at 2 to 8 Hz

frequency (Table 2). Temperature is measured with a reported accuracy of ± 0.2 K (± 0.3 K) for the

Pt 1000 (Sensirion) sensor. The temperature sensor specifications indicate a time lag of 3 to 30 s

but comparison between the two temperature sensors indicate that both have a similar lag of 2 to

5 s. The sensor lag appears as an offset between ascending and descending temperature profiles

during flights with continuous spiral ascent and descent flight patterns as shown in Cassano (2014)

and discussed in Section 4. As a result of this sensor lag most of the SUMO flights described in

this paper used stepped ascent or descent profiles (described in more detail below). Each step in

these ascent / descent patterns occurred over roughly 65 s so the temperature sensor lag becomes

unimportant as it is much shorter than the orbit time at each height. The reported sensor time

constant for relative humidity measurements, made with the Sensirion SHT 75 sensor, is

approximately 8 s although at temperatures well below 0°C we found that the humidity data was

largely unusable due to very long time lags. The Mayer et al. (2012) "no flow sensor" method was

used to estimate wind speed and direction by evaluating differences in ground speed throughout a

circular flight path and is described in greater detail in section 3.

**2.2 Flight Strategy**



A total of 116 SUMO sUAS flights took place between January 2012 and June 2017 at several locations in Antarctica (Fig. 1). Many of these flights took place over permanent ice shelf locations with 8 flights at Williams Field, 39 flights at the Pegasus ice runway and 36 flights at the Tall Tower AWS site (Wille et al. 2017). Pegasus runway and Williams Field are within 20 km of

the main United States Antarctic Program research base, McMurdo Station, and Ross Island, which has a maximum elevation in excess of 4000 m. The Tall Tower AWS flights took place approximately 160 km south southeast of McMurdo Station in a region of almost completely flat permanent ice with no major topographic features within 85 km of the site. Other continental flights occurred near Lake Vanda in the ice free McMurdo Dry Valleys (14 flights). Finally, several flights

occurred over sea ice in the western Ross Sea as part of the PIPERS cruise (19 flights; Ackley et al. 2020). To avoid issues related with aircraft icing all flights occurred in cloud free conditions, over the altitude range of the flight, although clouds above the maximum flight level were present for some flights. Table A1 lists the date and time, location and maximum altitude for each flight.

Additional Antarctic SUMO flights had been planned to take place after the PIPERS cruise

in 2017, but challenges in attempting to schedule a mid-winter campaign as well as the recent Antarctic field work restrictions due to COVID-19, resulted in the 2017 flights being the last SUMO flights conducted in Antarctica. As such, this paper provides a description of all Antarctic SUMO UAS flights that have been, or will be, conducted by our research group. Future sUAS flights in Antarctica, led by our group, will use the DataHawk2 sUAS, developed at the University of Colorado

- Boulder (Lawrence and Balsley 2013).

The primary scientific objective for all of our Antarctic SUMO flights was to obtain profiles of the atmospheric state of the boundary layer. During a given flight day SUMO boundary layer profile flights would occur every hour to several hours, although other factors, including weather and logistics, could limit the frequency and number of flights that could be performed. Changes in

the atmospheric thermodynamic state between pairs of SUMO profile flights allowed for estimation of the surface turbulent fluxes, based on state changes within the boundary (Bonin et al. 2013), as well as estimates of large-scale advective changes, based on state changes above the boundary layer. The relatively high temporal resolution atmospheric profiles observed by the SUMO were



also used to assess Antarctic Mesoscale Prediction System (Powers et al. 2012) operational

weather forecasts ( Wille et al. 2017).

A typical flight started with the SUMO manually or semi-autonomously controlled by the remote control pilot. Immediately after launch the remote control pilot would ensure that the SUMO sUAS was performing as expected and would then switch the aircraft to the fully autonomous flight mode. The aircraft would then climb to a specified height (usually 50 m agl) and orbit until instructed,

by the ground control pilot via the ground control software, to begin the profiling portion of the flight. A stepped spiral ascent / descent flight pattern was usually used with the aircraft set to orbit at several different heights below a maximum altitude of 1000 m agl. Some flights were conducted with a continuous spiral ascent followed by a continuous spiral descent, although as discussed above this flight pattern resulted in noticeable artifacts due to sensor lag. For both the stepped and

continuous profiles the spiral diameter was approximately 250 m. The SUMO would complete two circular orbits, in approximately 65 s, at each specified height in the stepped ascent / descent profile before climbing or descending to the next fixed height orbit location. Once the profiling was completed the aircraft would return to a height of approximately 50 m and orbit until the remote control pilot took control of the aircraft to land it in either manual or semi-autonomous mode.

The boundary layer depth in the Antarctic can vary from 10s of m or less during strongly stable, light wind conditions in winter to more than 1000 m in summer (King and Turner 1997). The maximum SUMO spiral profile heights ranged from 89 m to 1371 m agl (Table A1) and sampled the full depth of the boundary layer for all but a few flights. The continuous spiral ascent and descent flight pattern, up to 1000 m agl, usually took about 10 min to complete and two to three profiles

could be completed in a single flight. For the stepped ascent or descent profile flight patterns it was usually possible to complete 18 fixed height orbits during a 30 minute flight.

**3. Data processing and quality control**

Data from the SUMO sUAS were logged using an onboard datalogger controlled through

the Paparazzi autopilot software. This data was telemetered, via 2.4 GHz radio modem, to a laptop computer running the Paparazzi ground control software and also logged to an onboard SD





memory card. The data in the telemetered and SD data streams were identical, other than a reduced temporal resolution and occasional gaps in the record in the telemetered data. The SD data stream was used as the source data for the archived data except for cases where the SD data

was not available due to memory card failure or other issues.

The data recorded by the SUMO, in both the telemetered and SD data stream, is written sequentially with each record marked with the elapsed time since the SUMO was powered on. Each data record reported a single variable – aircraft status, navigation information or data from one of the scientific instruments. This data was stored in text log files on the ground station

computer and the onboard SD card.

Following each flight the SUMO SD, or telemetered, log files were processed into a comma separated text file. A header was written to this file that listed the flight location, sUAS pilots and start date and time of the flight. Each subsequent line of the file listed the date and time, elapsed time since SUMO power on, location and meteorological data (Table A2) at the same temporal

frequency as the original SUMO data file. The date and time variables were calculated from the date and time on the ground station laptop when the SUMO was powered on and the elapsed time reported for each record in the data file. Since each time period reported in the SUMO data file listed a single variable all other variables were flagged as missing values (-9999) for that time period. These data files were named yy_mm_dd__HH_MM_SS_SD.txt, where yy is year, mm is

month, dd is day, HH is hour, MM is minute and SS is second of the SUMO power on time in UTC. The _SD indicates that the data is from an SD file. If the data came from a telemetered SUMO log file the _SD was omitted in the filename.

A second data processing step linearly interpolated all variables in time to replace the missing data values and provide data for all variables at each time step in the data file. No

interpolation was performed before a given variable was first reported in the log file so these records retain missing data values. This data was written to files with the same naming convention as above but with _interpolation added to the filename before the .txt filename extension to indicate the temporal interpolation was applied to the original data.





The interpolated data was then used to calculate averages from each constant altitude
orbit completed during the flight and vertical bin averages. As described above, most SUMO flights
were conducted such that the sUAS orbited at multiple fixed heights over the flight altitude range.
These fixed height orbits helped address sensor time lag issues and also allowed the Mayer et al.
(2012) "no flow sensor" algorithm to estimate winds at a constant altitude. This processing step
identified flight segments that remain within ±8 m of all other points in the segment and for all
segments when the SUMO's position on the circular orbit passed through a full 360° (i.e. one
complete circular orbit). For each flight segment identified in this way the wind speed and direction
was estimated following Mayer et al. (2012) and the average altitude, temperature, relative humidity
and pressure was calculated. The wind speed was estimated based on the difference between the
minimum and maximum GPS ground speed recorded during the circular orbit. The wind direction
was calculated based on the orbit heading at the location of the minimum and maximum GPS
speed, with both directions reported in the archived data file. The start and end time, heading and
altitude, as well as elapsed time, on the constant height orbit were also reported in the archived
data file named yy_mm_dd_HH_MM_SS_SD_const_alt.txt. All data stored in the constant altitude
archived data file are listed in Table A2.

Vertical bin averages were calculated for each 5 m altitude bin over the full altitude range
of each SUMO flight. For each vertical altitude bin the average, standard deviation and number of
observations in the bin was calculated. These values were calculated from all data during the flight
as well as from data from the ascent-only and descent-only portions of the flight. This data is stored
in files named yy_mm_dd__HH_MM_SS_SD_vert_avg.txt (Table A2).

Data from each flight, in the interpolated data file as well as from the vertical bin average
and constant altitude orbit data files was visually inspected. The following issues were observed,
but no data was removed from the archived data files. For each flight, there is a short period, of a
few seconds, when the temperature sensors adjust to the ambient conditions immediately after
take-off. As noted in section 2.1 there is a noticeable sensor lag evident in the temperature and
humidity measurements. The sensor lag is short, of the order of a few seconds, for the temperature
measurements but is much longer for the humidity observations. These lags are obvious when





comparing observations from the ascent and descent portions of the flight (Fig. 3). Since the original time resolution data is provided users of this data can apply whatever corrections are needed for their application to account for the sensor lag, but no corrections were applied to the archived data.


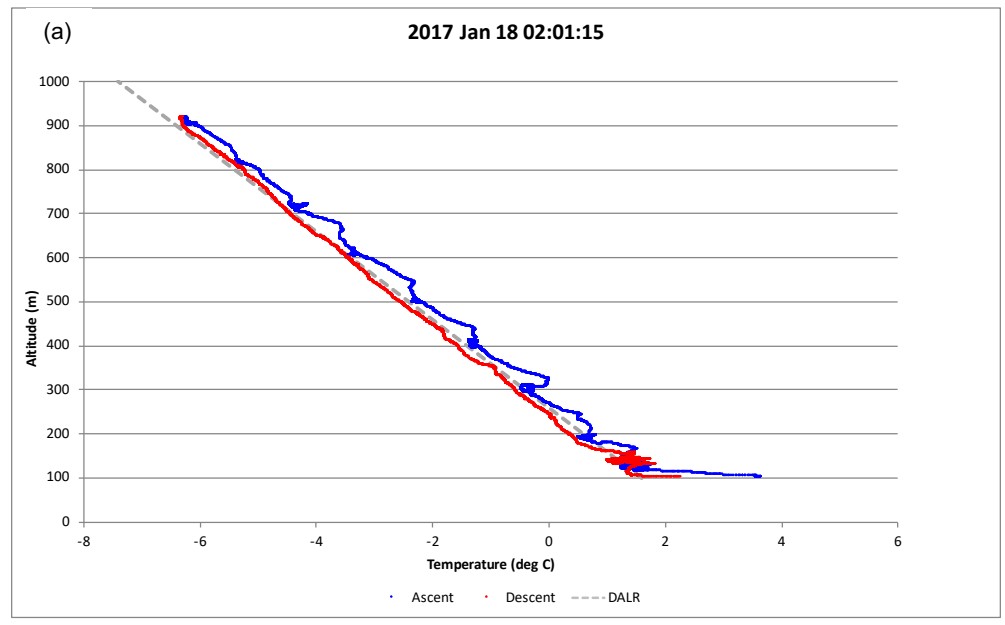



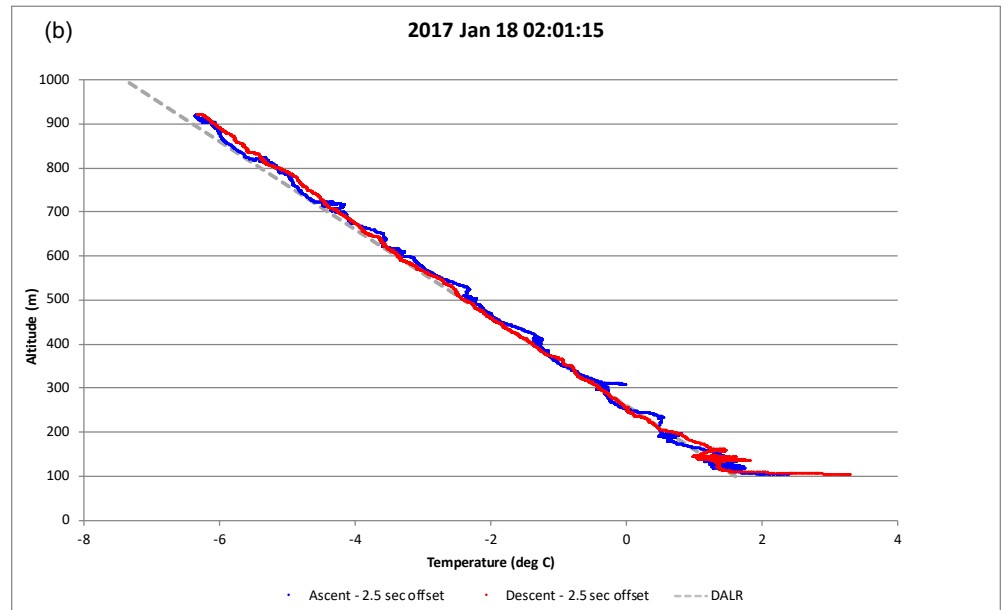

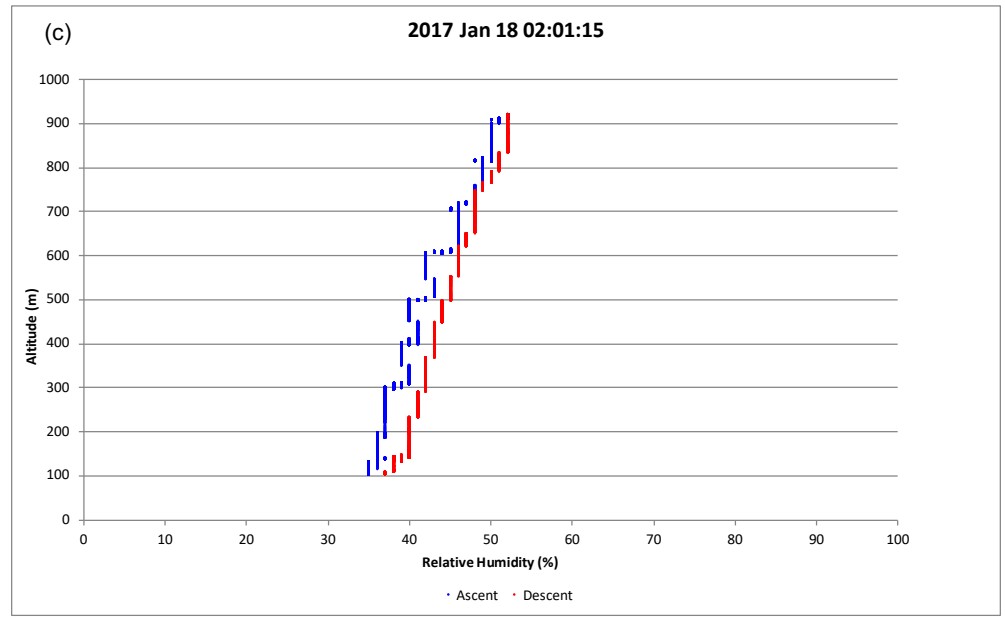


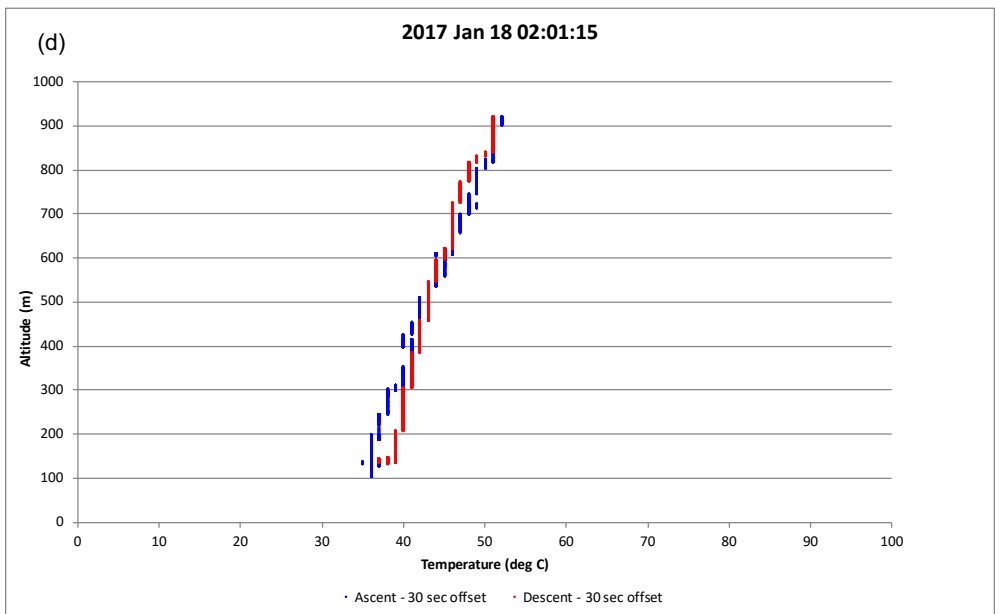

Figure 3. Temperature (a and b) and relative humidity (c and d) profiles observed by the SUMO sUAS in the Wright Valley, Antarctica at 0201 UTC 18 January 2017. The original time interpolated data is plotted in panels a and c and data with a time lag of 2.5 s and 30 s is plotted in panels b and d, respectively.

Figure 3 illustrates the sensor lag for temperature and humidity measurements for a SUMO flight conducted at 0201 UTC 18 January 2017 in the Wright Valley. During this flight a deep, convective boundary layer was present. Figure 3a shows the temperature observed during the stepped ascent (blue) and spiral descent (red) portions of this flight. The ascent portion of the flight is consistently warmer than the descent and given the decreasing temperature with height on this day, is consistent with a short time lag in the temperature measured by the SHT sensor. By applying a 2.5 s offset to the temperature, relative to the height, the ascent and descent profiles more closely match (Fig. 3b). This time lag is consistent with results presented by Cassano (2014). The humidity observations also exhibit a time lag (Fig. 3c). Here, humidity increases with height and the time lag results in the ascending observations being shifted to slightly lower relative humidity values, at a



given height, compared to the observations during the descent. Applying a 30 s time shift to the

humidity measurements results in much closer correspondence between the humidity profiles

observed during the SUMO ascent and descent (Fig. 3d), although does not fully remove the

discrepancies between the ascent and descent profiles as seen for temperature (Fig. 3b).

**4. Examples of Observed Features**

295        With SUMO flights being conducted throughout the full annual cycle and over a variety of

surface conditions across the Antarctic continent a wide range of boundary layer stability, depth

and evolution have been observed. Some examples of the variety of boundary layer states

observed by the SUMO UAS are given below.

        Eleven SUMO flights were conducted at the Ross Ice Shelf Tall Tower site over a 14.5 h

period from 1916 UTC 20 January 2014 to 0944 UTC 21 January 2014 (Fig. 4). The timing between

most of these flights was ~1.5 hours which provided a detailed depiction of the temporal evolution

of the boundary layer and free atmosphere above.

        From 1916 UTC 20 January to 0204 UTC 21 January a shallow, well mixed boundary was

observed, with a dry adiabatic temperature profile up to a maximum height of 250 m. After 0204

UTC the boundary layer transitioned to a weakly stable temperature profile. The temperature in the

boundary layer initially cooled (1916 to 2144 UTC 20 January) ~1 K and then warmed ~5 K by 0731

UTC 21 January before cooling ~2 K over the next  approximately 2 hours.

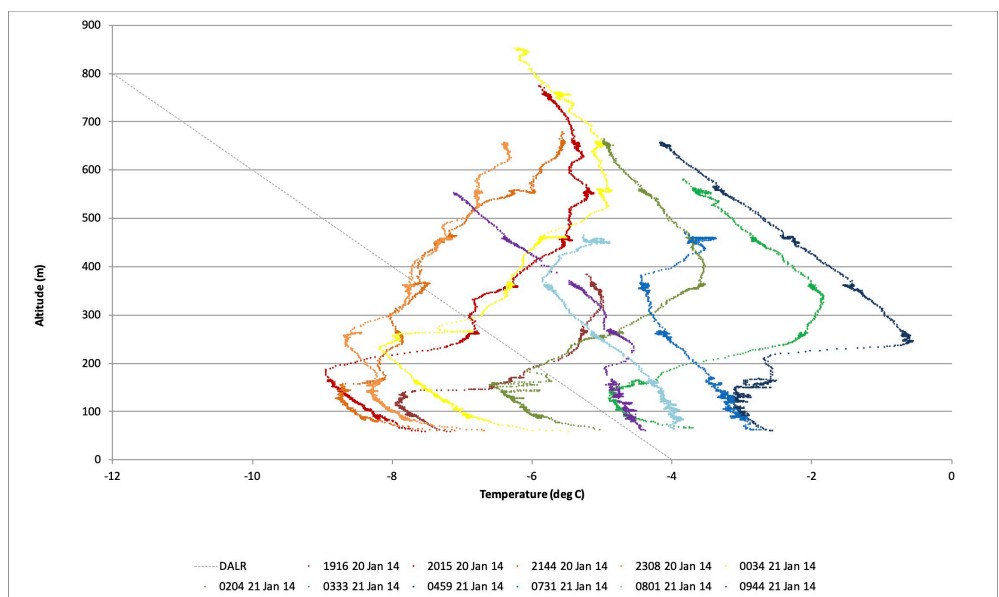

Figure 4: SUMO observed temperature profiles at the Ross Ice Shelf Tall Tower AWS

site from 1916 UTC 20 January 2014 to 0944 UTC 21 January 2014. Temperature

profiles are plotted as colored lines (20 Jan 1916 UTC – dark red, 20 Jan 2015 UTC –

red; 20 Jan 2144 UTC – dark orange; 20 Jan 2308 UTC – orange; 21 Jan 0034 UTC –

yellow; 21 Jan 0204 UTC – dark green; 21 Jan 0333 UTC – green; 21 Jan 0459 UTC –

dark blue; 21 Jan 0731 UTC – blue; 21 Jan 0801 UTC – light blue; 21 Jan 0944 UTC –

purple). A dry adiabatic temperature profile is shown for reference as a gray dotted line.

The temperature above the boundary layer was not constant during the 14.5 h sampling

period from 20-21 January 2014. Considering the temperature at 300 m, the temperature cooled

~3 K from 1916 to 2308 UTC 20 January. This cooling was greater than what was observed in the

boundary layer during this same time period and suggests that large-scale advective cooling, as

inferred from the cooling aloft, was offset by an upward sensible heat flux in the boundary layer,

consistent with the convective conditions observed at this time. Over the next four flights (0034 to

0459 UTC 21 January) the temperature at 300 m warmed almost 8 K, while the boundary layer

temperature did not warm as much. This suggests that large-scale warm advection occurring at

this time was offset by either radiative cooling or a downward sensible heat flux in the boundary

layer, consistent with the transition from a convective to slightly stable boundary layer during this time period. During the final three flights (0731 to 0944 UTC 21 January) cooling of ~1 K was observed aloft and in the boundary layer.

330         Contrasting the summer conditions seen in Fig. 4, Fig. 5 shows temperature profiles observed during 11-12 September 2016 at the Pegasus runway at the end of the Antarctic winter. Six SUMO flights were completed between 1426 UTC 11 September and 1658 UTC 12 September 2016. The surface temperature during these flights was near -40°C, which is the lower observation limit of the SHT temperature sensor. Unlike the temperature profiles shown in Fig. 4, the

temperature profiles observed during the 24+ hour period from 11-12 September 2016 (Fig. 5) showed a remarkable lack of temporal variability, exhibiting nearly steady state conditions. During this time a very strong, shallow surface inversion was present with the temperature warming ~7 K in the lowest 50 m of the sounding and then warming another several K up to 200 m.

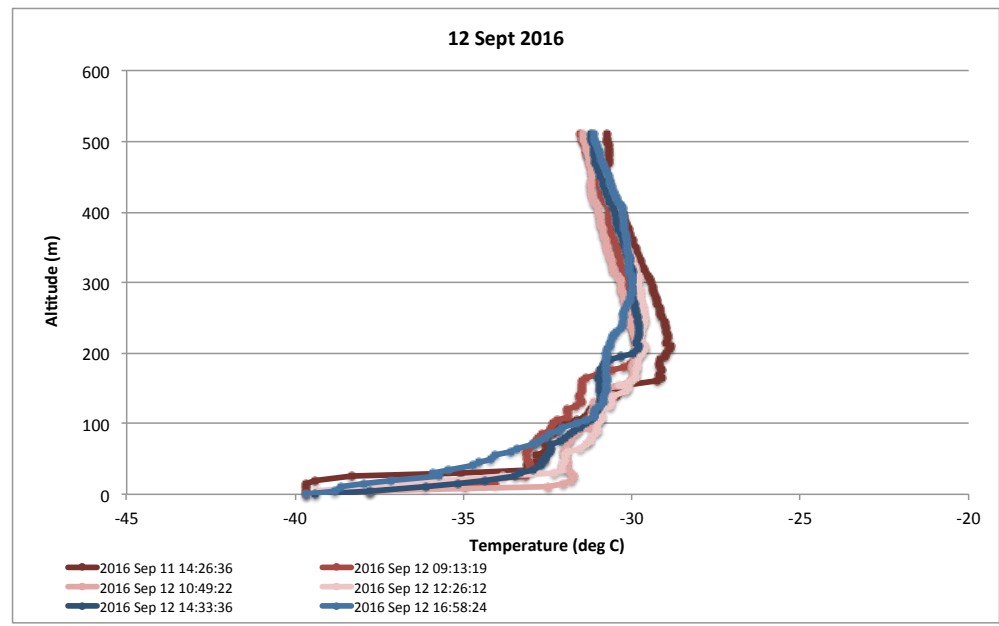


Figure 5: SUMO observed temperature profiles at the Pegasus Ice Runway from 1426 UTC 11 September 2016 to 1658 UTC 12 September 2016. Temperature profiles are plotted as colored lines (11 Sept 1426 UTC – dark red, 12 Sept 0913 UTC – red; 12 Sept 1049 UTC





– pink; 12 Sept 1226 UTC – light pink; 12 Sept 1433 UTC – dark blue; 12 Sept 1658 UTC

345          – blue).

During the austral summer of 2017 SUMO flights were conducted in the Wright Valley, near

Lake Vanda, at one of the few permanently ice free locations on the Antarctic continent. The

temperature profiles observed from 16-18 January 2017 (Fig. 6) differ markedly from the profiles

shown in Figs. 4 and 5. The boundary layer observed in the Wright Valley was a deep convective

boundary layer with a surface temperature near or just above 0°C. From late morning to mid-

afternoon local time (1933 UTC 17 Jan 2017 to 0201 UTC 18 Jan 2017) the boundary layer warmed

~2 K and deepened from 200 m to more than 800 m, with a dry adiabatic lapse rate extending

beyond the top flight altitude of the SUMO at 0057 and 0201 UTC 18 January 2017.


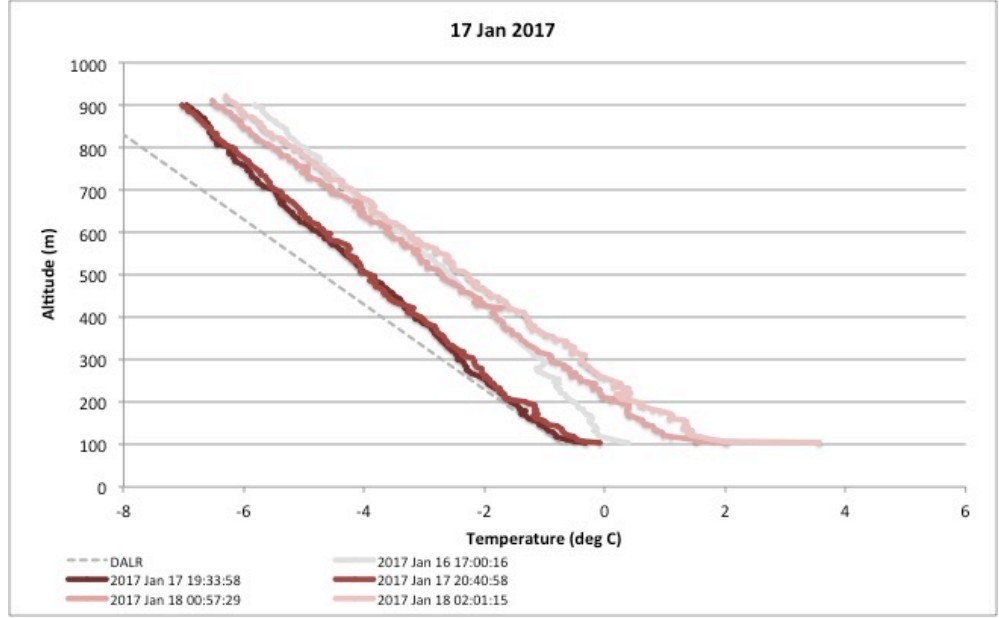

Figure 6: SUMO observed temperature profiles near Lake Vanda in the Wright Valley from

1700 UTC 16 January 2017 to 0201 UTC 18 January 2017. Temperature profiles are

plotted as colored lines (16 Jan 1700 UTC – gray, 17 Jan 1933 UTC – dark red; 17 Jan

2041 UTC – red; 18 Jan 0057 UTC – pink; 18 Jan 0201 UTC – light pink).



### 5. Data Availability

The SUMO sUAS data described in this paper can be retrieved from the United States Antarctic Program Data Center (https://www.usap-dc.org). The data for all flights conducted on the

continent (Williams Field, Pegasus runway, Tall Tower AWS site and Wright Valley) are available at https://www.usap-dc.org/view/dataset/601054 (Cassano 2017; https://doi.org/10.15784/601054) and data from the PIPERS cruise, in the Ross Sea, are available at https://www.usap-dc.org/view/dataset/601191 (Cassano 2019; https://doi.org/10.15784/601191). The data is archived in annual zip files that contain comma delimited text files of the data at its original and

interpolated time resolution and as vertical bin averaged and constant altitude data. Each data file contains a header that lists the flight location name, latitude, longitude, start date and time (UTC) of the flight and the sUAS pilots. A final header line lists the data type and units contained in each comma separated column in the remainder of the file.

**6. Summary**

Between January 2012 and June 2017 a small unmanned aerial system (UAS), known as the Small Unmanned Meteorological Observer (SUMO), was used to observe the temperature, humidity, pressure and wind in and above the Antarctic atmospheric boundary layer. During 6 Antarctic field campaigns 116 SUMO flights were completed. Flights over ice shelf locations

occurred at Williams Field and the Pegasus runway (2 field campaigns), both located within 20 km of McMurdo Station and Ross Island, and at the Tall Tower AWS site located in the northwestern portion of the Ross Ice Shelf (Fig. 1). Flights also took place in the ice free Wright Valley, near Lake Vanda in a region of complex terrain and were performed over sea ice, in the western Ross Sea, as part of the PIPERS research cruise (Ackley et al. 2020). These flights took place during all

seasons with most of the flights conducted during the austral summer (January) and late winter / early spring (September). The flights observed the full depth of the atmospheric boundary layer and a portion of the free atmosphere above, with the SUMO flying a spiral ascent and descent flight path. A wide variety of boundary layer states were observed including very shallow, strongly stable



conditions during the Antarctic winter (Fig. 5) and deep, convective conditions over ice free
locations in the summer (Fig. 6).

Data from these flights was processed from the native SUMO log files into comma separated text files at the original and an interpolated time resolution. Additional data processing, of the interpolated time resolution data, created vertical bin averaged data and averages over constant altitude SUMO orbits. Errors noted in the data include short periods of time, at the start of
the flight, when the meteorological sensors equilibrate with the ambient atmospheric conditions and time lags in the temperature (~2.5 s) and humidity (30 or more s) measurements.

The Antarctic atmospheric boundary layer data collected by the SUMO sUAS, described in this paper, are freely available from the United States Antarctic Program Data Center (https://www.usap-dc.org). The data for all flights conducted on the continent (Williams Field,
Pegasus ice runway, Tall Tower AWS site and Wright Valley) are available at https://www.usap-dc.org/view/dataset/601054 (Cassano 2017; https://doi.org/10.15784/601054) and data from the PIPERS cruise Ross Sea flights, are available at https://www.usap-dc.org/view/dataset/601191 (Cassano 2019; https://doi.org/10.15784/601191).

**Author Contributions.** J.J. Cassano was the lead investigator for the field campaigns described here and piloted most of the SUMO sUAS flights. He post-processed and quality controlled all of the data. M.A. Nigro served as the ground control pilot for the 2014 Ross Ice Shelf SUMO sUAS flights and created some of the code used to post-process the SUMO data. M.W. Seefeldt served as the ground control pilot for the 2016 Pegasus SUMO sUAS flights. M. Katurji assisted with the
2017 Wright Valley SUMO UAS flights. K. Schick and G. Williams conducted the SUMO flights during the PIPERS cruise in 2017. A. DuVivier assisted with SUMO flights at Williams Field in January 2012.

**Acknowledgements.** This work was supported by NSF grants ANT 0943952 and ANT 1245737
and by funding from the New Zealand Ministry of Business, Innovation and Employment (UOWX1401, the Dry Valley Ecosystem Resilience [DryVER] project) and through the Antarctic



Science Platform (ANTA1801). The authors thank the United States Antarctic Program and Antarctica New Zealand personnel for their help during the field campaigns described in this paper. The authors also wish to thank Shelley Knuth for her help with flights at Pegasus Runway in

September 2012 and Martin Müller and Christian Lindenberg for developing the SUMO sUAS used for this research and providing the training to J.J. Cassano to operate the SUMO and ground station software.

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



**Appendix A**

Table A1: UTC date and time (yymmdd_HHMM format; yy – year; mm – month; dd – day; HH – hour; MM - minute), location and maximum altitude above ground level (m agl) for all SUMO flights.

| Date and time (UTC) | Location | Maximum altitude (m agl) |
|---|---|---|
| 20120114_1810 | Williams Field | 1002 |
| 20120121_1237 | Williams Field | 1005 |
| 20120121_1659 | Williams Field | 999 |
| 20120121_1842 | Williams Field | 1008 |
| 20120128_0022 | Williams Field | 713 |
| 20120128_1254 | Williams Field | 1013 |
| 20120128_1641 | Williams Field | 1025 |
| 20120128_1910 | Williams Field | 1371 |
| 20120912_2031 | Pegasus Runway | 1014 |
| 20120915_1947 | Pegasus Runway | 886 |
| 20120915_2103 | Pegasus Runway | 969 |
| 20120915_2230 | Pegasus Runway | 1015 |
| 20120916_0255 | Pegasus Runway | 776 |
| 20120923_2004 | Pegasus Runway | 510 |
| 20140116_0031 | Tall Tower | 418 |
| 20140116_0308 | Tall Tower | 540 |
| 20140116_2336 | Tall Tower | 526 |
| 20140117_0027 | Tall Tower | 256 |
| 20140117_0129 | Tall Tower | 553 |
| 20140117_0310 | Tall Tower | 654 |
| 20140118_2226 | Tall Tower | 621 |
| 20140118_2345 | Tall Tower | 717 |
| 20140119_0119 | Tall Tower | 802 |
| 20140119_0232 | Tall Tower | 614 |
| 20140119_0404 | Tall Tower | 398 |
| 20140120_0824 | Tall Tower | 151 |
| 20140120_1916 | Tall Tower | 321 |
| 20140120_2015 | Tall Tower | 714 |
| 20140120_2308 | Tall Tower | 599 |
| 20140121_0035 | Tall Tower | 798 |
| 20140121_0204 | Tall Tower | 600 |
| 20140121_0333 | Tall Tower | 514 |
| 20140121_0459 | Tall Tower | 595 |
| 20140121_0631 | Tall Tower | 416 |
| 20140121_0802 | Tall Tower | 400 |
| 20140121_0945 | Tall Tower | 490 |
| 20140122_0025 | Tall Tower | 283 |
| 20140122_2111 | Tall Tower | 687 |
| 20140122_2324 | Tall Tower | 621 |
| 20140123_0116 | Tall Tower | 498 |
| 20140123_0419 | Tall Tower | 690 |
| 20140123_0620 | Tall Tower | 502 |
| 20140123_2024 | Tall Tower | 511 |
| 20140123_2150 | Tall Tower | 693 |
| 20140123_2318 | Tall Tower | 610 |





| | | |
|---|---|---|
| 20140124_0051 | Tall Tower | 600 |
| 20140124_0225 | Tall Tower | 720 |
| 20140124_0351 | Tall Tower | 403 |
| 20140124_0510 | Tall Tower | 196 |
| 20140124_0837 | Tall Tower | 696 |
| 20160907_0151 | Pegasus Runway | 507 |
| 20160907_0333 | Pegasus Runway | 173 |
| 20160907_0527 | Pegasus Runway | 519 |
| 20160907_2202 | Pegasus Runway | 513 |
| 20160907_2334 | Pegasus Runway | 507 |
| 20160908_0256 | Pegasus Runway | 524 |
| 20160908_0445 | Pegasus Runway | 506 |
| 20160910_0335 | Pegasus Runway | 498 |
| 20160910_0524 | Pegasus Runway | 506 |
| 20160910_0708 | Pegasus Runway | 507 |
| 20160910_0854 | Pegasus Runway | 507 |
| 20160910_1043 | Pegasus Runway | 509 |
| 20160911_0909 | Pegasus Runway | 503 |
| 20160911_1055 | Pegasus Runway | 516 |
| 20160911_1243 | Pegasus Runway | 512 |
| 20160911_1427 | Pegasus Runway | 495 |
| 20160912_0913 | Pegasus Runway | 511 |
| 20160912_1049 | Pegasus Runway | 508 |
| 20160912_1226 | Pegasus Runway | 506 |
| 20160912_1434 | Pegasus Runway | 509 |
| 20160912_1658 | Pegasus Runway | 511 |
| 20160917_0405 | Pegasus Runway | 512 |
| 20160917_0541 | Pegasus Runway | 505 |
| 20160919_0612 | Pegasus Runway | 508 |
| 20160919_0809 | Pegasus Runway | 506 |
| 20160919_2125 | Pegasus Runway | 514 |
| 20160924_0115 | Pegasus Runway | 515 |
| 20160924_0318 | Pegasus Runway | 507 |
| 20160924_0718 | Pegasus Runway | 506 |
| 20160927_0127 | Pegasus Runway | 507 |
| 20160928_0114 | Pegasus Runway | 505 |
| 20160928_0724 | Pegasus Runway | 503 |
| 20160928_1601 | Pegasus Runway | 494 |
| 20170112_2135 | Lake Vanda | 797 |
| 20170114_2119 | Lake Vanda | 792 |
| 20170115_0325 | Lake Vanda | 797 |
| 20170115_1205 | Lake Vanda | 784 |
| 20170115_1309 | Lake Vanda | 796 |
| 20170116_1123 | Lake Vanda | 800 |
| 20170116_1321 | Lake Vanda | 813 |
| 20170116_1517 | Lake Vanda | 797 |
| 20170116_1559 | Lake Vanda | 805 |
| 20170116_1700 | Lake Vanda | 795 |
| 20170117_1934 | Lake Vanda | 798 |
| 20170117_2041 | Lake Vanda | 797 |
| 20170118_0057 | Lake Vanda | 801 |
| 20170118_0201 | Lake Vanda | 818 |
| 20170425_0434 | Ice Station 1 | 89 |





| 20170425_0550 | Ice Station 1 | 194 |
|---|---|---|
| 20170425_0850 | Ice Station 1 | 501 |
| 20170426_0234 | Ice Station 2 | 383 |
| 20170513_2307 | Ice Station 3 | 256 |
| 20170514_0327 | Ice Station 3 | 398 |
| 20170514_0912 | Ice Station 3 | 405 |
| 20170527_0352 | Ice Station 7 | 297 |
| 20170527_0444 | Ice Station 7 | 303 |
| 20170527_2335 | Ice Station 8 | 599 |
| 20170528_0152 | Ice Station 8 | 119 |
| 20170529_0836 | Ice Station 9 | 216 |
| 20170529_1050 | Ice Station 9 | 902 |
| 20170529_1227 | Ice Station 9 | 897 |
| 20170531_0206 | Ice Station 10 | 907 |
| 20170531_0511 | Ice Station 10 | 908 |
| 20170531_2046 | Ice Station 11 | 154 |
| 20170602_0314 | Ice Station 12 | 898 |
| 20170602_0618 | Ice Station 12 | 808 |




Table A2. Post-processed SUMO data files and data archived in each type of post-processed file.

| Post-processed SUMO data file[*] | Variables stored in file |
|---|---|
| yy_mm_dd__HH_MM_SS_SD.txt | UTC year, month, day, hour, minute, second |
| yy_mm_dd__HH_MM_SS_SD_interpolation.txt | Elapsed time since SUMO power on (sec) |
| | Easting, northing, altitude (m) |
| | GPS speed (m s$^{-1}$) |
| | Relative humidity from SHT sensor (%) |
| | Temperature from SHT and Pt sensors (deg C) |
| | Pressure from VTI sensor (mb) |
| | Downward facing infrared temperature (deg C) |
| | Original observation flag |
| yy_mm_dd__HH_MM_SS_SD_const_alt.txt | Count – sequential counter identifying each constant altitude orbit |
| | UTC year, month, day, hour, min, sec |
| | Time (sec) |
| | Altitude (m) |
| | Pressure (mb) |
| | Temperature (SHT, Pt and IR) (deg C) |
| | RH (%) |
| | Wind speed (m s$^{-1}$) |
| | Wind direction (deg) (two estimates) |
| | Constant altitude orbit time start and end (sec) |
| | Total time for constant altitude orbit (sec) |
| | SUMO heading at start and end of constant altitude orbit (deg) |
| | Summed change in heading over orbit (deg) |
| | Start and end altitude for constant altitude orbit (m) |
| | GPS minimum and maximum speed on constant altitude orbit (m s$^{-1}$) |
| | Heading at mimimum and maximum GPS speed (deg) |
| yy_mm_dd__HH_MM_SS_SD_vert_avg.txt | Bin altitude (m) |
| | For each of the following variables the bin average, standard deviation and number of observations used to calculate the bin average are reported for all observations |





| | during the flight and for all ascent-only and all descent-only observations. |
| --- | --- |
| | Altitude (m) |
| | RH (%) |
| | Temperature (SHT, Pt and IR) (deg C) |
| | Pressure (mb) |

* _SD portion of filename is omitted if data came from telemetry data stream rather than SD data stream.