# Peer review of "Antarctic atmospheric boundary layer observations with the Small Unmanned Meteorological Observer (SUMO)"

_Earth System Science Data, 2020_

## Referee Comment (RC1) · Anonymous Referee #1 · 21 Dec 2020

The manuscript presents a very clear and careful documentation of the SUMO UAS observations on the lower tropospere collected from the Antarctic during 2012 to 2017. The data set is unique and will probably receive a lot of attention. It was a pleasure to read the manuscript. I suggest acceptance of the manuscript subject to minor revisiosn specified below.

Line 29. Acronyme sUAS should be defined.

Figure 1. Increase font size for the texts below plots (a) and (b).

Table 1. Are you sure than SUMO can fly as fast as 42 m/s? Or is it only possible in strong tail wind?

Table 2. Why "not applicable" is given for the accuracy of pressure and for the sensor time constant for pressure and wind?

Line 289. If 30 s is found good here, why 8 s is given for sensor time constant in Table 2?

Figure 4. Below the plot, very small dots are used to identify the dates. Increase the dot size for better readability.

Lines 318-328. The discussion is interesting but remains speculative, as no attempts are made to estimate the magnitide of advective heating / cooling. Also subsidence heating may play a role. Consider if it is better to drop these discussions or make some more effort to provide quantitative estimates (based e.g. on reanalysis fields) for the roles of horizontal heat advection and subsidence heating.

---

## Referee Comment (RC2) · Anonymous Referee #2 · 8 Jan 2021

This is a very nice, well written and informative summary and data collection of atmospheric profile measurements over Antarctica with the unmanned aerial system SUMO (Small Unmanned Meteorological Observer) that can in my opinion be accepted for publication with some minor changes. My main criticism is the formatting of the figures presenting the data (Figs 3-6) that are not really high scientific standard and should be improved. They are in addition partially hard to read e.g. due to rather small labels. I also strongly suggest to include readable legends, which would also make it unnecessary to have excessive figure captions listing different colors that further hampers the figure readability.

[Figure]

Some other minor comments that could be worth considering: Line 76: here would be a suitable location for the general and primary SUMO reference Reuder, J., P. Brisset, M. Jonassen, M. Müller, and S. Mayer, The Small Unmanned Meteorological Observer SUMO: A new tool for atmospheric boundary layer research, Meteorologische Zeitschrift, 18, 2, 141-147, 2009

Line 176: there is another comprehensive paper on the Bonin method applied on SUMO measurements that also includes a few stable situations Båserud, L., J. Reuder, M. O. Jonassen, T. A. Bonin, P. B. Chilson, M. A. Jimenez, and P. Durand, Potential and limitations in estimating sensible heat flux profiles from consecutive temperature profiling by RPAS, Boundary-Layer Meteorology, 174(1), 145-177, DOI: 10.1007/s10546-019-00478-9, 2020

Line 186: it is not completely clear to me if the descent was always continuous or if there were also cases of stepped descent?

Lines 250-254: has this bin averaging also been applied for the stepped profiles? If so, does this make sense as the constant height circles will lead to very heterogeneous distribution of number of measurements per bin?

Figure 3: I could not find that "DALR" in the legend was defined before; maybe just mention "dry adiabatic lapse rate" in the caption?

Figure 3d: x-axis label should read "relative humidity" not "temperature"

References: several inconsistencies in the formatting of journal names (abbreviated/not abbreviated)

Table A1: I suggest to include information on: - Stepped/continuous profiling - Flight duration - Number of profiles performed during one flight (if applicable)

---

## Author Comment (AC3) · 30 Jan 2021

Response to anonymous referee comments

The authors thank the two anonymous referees for taking the time to review our manuscript and for their helpful comments, which have improved the manuscript. Each referee comment is given below followed by our response to the comment.

Anonymous referee #1

Referee:The manuscript presents a very clear and careful documentation of the SUMO UAS observations on the lower trospere collected from the Antarctic during 2012 to

2017. The data set is unique and will probably receive a lot of attention. It was a pleasure to read the manuscript. I suggest acceptance of the manuscript subject to minor revisiosn specified below.

Thank you for your positive comment about our manuscript.

Referee:Line 29. Acronyme sUAS should be defined.

Thank you for pointing out this oversight on our part. We now define the acronym sUAS when the term small unmanned aerial system is first used in the abstract and again when this term is first used in the main manuscript text.

Referee:Figure 1. Increase font size for the texts below plots (a) and (b).

Unfortunately, we are unable to modify the size of the font on these figures. These figures were created for us by the Polar Geospatial Center at the University of Minnesota and we do not have access to the original files for modifying the font size. For the revised manuscript we will upload the original resolution figures so that when viewed at full size all text will be easily legible.

Referee: Table 1. Are you sure than SUMO can fly as fast as 42 m/s? Or is it only possible in strong tail wind?

The SUMO can be configured with a more powerful motor and smaller propeller to achieve a maximum speed of 42 m s-1 but as configured for our campaigns the maximum speed is 25 m s-1. We have updated the information in Table 1 to indicate the maximum speed as used for our Antarctic campaigns.

Referee:Table 2. Why "not applicable" is given for the accuracy of pressure and for the sensor time constant for pressure and wind?

In table 2 we now list the relative accuracy of 0.5 hPa, as given by the manufacturer, for the VTI SCP 1000 pressure sensor. The manufacturer has not provided a sensor time constant for the VTI SCP 1000 so we now indicate that this information is not available.

The "no flow" wind estimate is based on changes in GPS speed over a single circular orbit of the SUMO so the wind estimate is not applicable to a single point and thus a time constant is not applicable. We do indicate that it requires ∼30 s to acquire one wind estimate.

Referee:Line 289. If 30 s is found good here, why 8 s is given for sensor time constant in Table 2?

We have updated the discussion of the humidity profiles in Figure 3d to indicate that we suspect that the longer time constant we found for our humidity data may be related to the cold temperatures during our flights, compared to what the manufacturer likely used when characterizing this sensor.

Referee:Figure 4. Below the plot, very small dots are used to identify the dates. Increase the dot size for better readability.

We have increased the size of the symbols used to plot the temperature profiles in Figure 4 to improve the readability of this figure and legend.

Referee:Lines 318-328. The discussion is interesting but remains speculative, as no attempts are made to estimate the magnitide of advective heating / cooling. Also subsidence heating may play a role. Consider if it is better to drop these discussions or make some more effort to provide quantitative estimates (based e.g. on reanalysis fields) for the roles of horizontal heat advection and subsidence heating.

We agree that our discussion here is speculative. We prefer to leave this discussion in the manuscript to illustrate the types of thermodynamic processes which can be inferred from the SUMO temperature data. We have added the following text to indicate that this discussion is speculative and requires further analysis in the future. Here, we assume that the temperature change observed above the boundary layer is due to large-scale advective changes, but other processes such as adiabatic changes due to vertical motion or radiative heating or cooling may also contribute to the observed

temperature trends. Further analysis would be required to determine the relative role of each of these processes in altering the temperature aloft.

Anonymous referee #2

Referee:This is a very nice, well written and informative summary and data collection of atmospheric profile measurements over Antarctica with the unmanned aerial system SUMO (Small Unmanned Meteorological Observer) that can in my opinion be accepted for publication with some minor changes.

Thank you for your positive comment about our manuscript.

Referee:My main criticism is the formatting of the figures presenting the data (Figs 3-6) that are not really high scientific standard and should be improved. They are in addition partially hard to read e.g. due to rather small labels. I also strongly suggest to include readable legends, which would also make it unnecessary to have excessive figure captions listing different colors that further hampers the figure readability.

We have increased the font size for all text on these figures, including in the legend, and hope that they are now more legible. We have also removed the details regarding the colors used to plot each profile from the figure caption.

Referee:Some other minor comments that could be worth considering: Line 76: here would be a suitable location for the general and primary SUMO reference Reuder, J., P. Brisset, M. Jonassen, M. Müller, and S. Mayer, The Small Unmanned Meteorological Observer SUMO: A new tool for atmospheric boundary layer research, Meteorologische Zeitschrift, 18, 2, 141-147, 2009

Thank you for suggesting that we include this primary SUMO reference. We now cite this paper when we first introduce the SUMO UAS in the introduction of our manuscript.

Referee:Line 176: there is another comprehensive paper on the Bonin method applied on SUMO measurements that also includes a few stable situations Båserud, L., J. Reuder, M. O. Jonassen, T. A. Bonin, P. B. Chilson, M. A. Jimenez, and P. Durand, Potential and limitations in estimating sensible heat flux profiles from consecutive temperature pro- filing by RPAS, Boundary-Layer Meteorology, 174(1), 145-177, DOI: 10.1007/s10546- 019-00478-9, 2020

Thank you for drawing our attention to this paper. We now cite this along with the Bonin et al. (2013) paper.

Referee:Line 186: it is not completely clear to me if the descent was always continuous or if there were also cases of stepped descent?

For most flights a stepped ascent was followed by a continuous descent, although some flights had a stepped ascent followed by a stepped descent.

Referee:Lines 250-254: has this bin averaging also been applied for the stepped pro- files? If so, does this make sense as the constant height circles will lead to very het- erogeneous distribution of number of measurements per bin?

Yes, the bin averaging is applied for all flights, including the stepped profiles. As the reviewer has indicated this does result in a different number of observations in each bin, but because there was very little change in observed atmospheric state during each fixed height orbit we believe that all bin averaged values are comparable despite the varying number of observations in each bin.

Referee:Figure 3: I could not find that "DALR" in the legend was defined before; maybe just mention "dry adiabatic lapse rate" in the caption?

We now define DALR in the caption for Figure 3.

Referee:Figure 3d: x-axis label should read "relative humidity" not "temperature"

Thank you for noticing this mistake. We have corrected the axis label in Figure 3d.

Referee:References: several inconsistencies in the formatting of journal names (ab- breviated/not abbreviated)

We have reviewed all of the references to ensure that all journal names are no longer abbreviated.

Referee:Table A1: I suggest to include information on: - Stepped/continuous profiling - Flight duration - Number of profiles performed during one flight (if applicable)

Thank you for this suggestion. We have updated Table A1 to include the flight duration and the type of profile (stepped or continuous ascent or descent). Almost all flights consisted of one vertical profile, but we have noted flights when multiple profiles were flown.